# A Training Scheme for the Uncertain Neuromorphic Computing Chip Conference Submissions

## Abstract

Uncertainty is a very important feature of the intelligence and helps the brain become a flexible, creative and powerful intelligent system. The crossbar-based neuromorphic computing chips, in which the computing is mainly performed by analog circuits, have the uncertainty and can be used to imitate the brain. However, most of the current deep neural networks have not taken the uncertainty of the neuromorphic computing chip into consideration. Therefore, their performances on the neuromorphic computing chips are not as good as on the original platforms (CPUs/GPUs). In this work, we proposed the uncertainty adaptation training scheme (UATS) that tells the uncertainty to the neural network in the training process. The experimental results show that the neural networks can achieve comparable inference performances on the uncertain neuromorphic computing chip compared to the results on the original platforms, and much better than the performances without this training scheme.

## 1 Introduction

Uncertainty reasoning is the essence of human thinking activities and a key aspect of the intelligence. There are two kind of uncertainties in intelligent systems. One is the fuzziness, the other is the stochasticity. The fuzziness helps the brain deal with the real world efficiently by ignoring the enormous redundant information. When we try to distinguish a cat or a dog, we do not need to know the expressions and the number of the legs. Although such information can be easily captured by our visual system with a glance, it will be ignored for efficiency. The stochasticity endows the brain the creativity and enables us not always failed in an unfamiliar field. Our decisions may change when we do not sure. These characteristics are not available in most existing artificial intelligence (AI) systems, such as a classifier based on a deep neural network (DNN). The 32-bit or 64-bit floating numbers are used to describe the weights and activations. While some researchers found that the 8-bit integer is enough for many applications Banner et al. (2018); Wu et al. (2018). Moreover, after the training procedure, the results will be the same no matter how many times it performs, although the margin is very small and the answer is wrong. There are some methods to address these issues, such as the network quantization and the Bayesian network. In addition, the neuromorphic computing chip has provide a hardware approach to supplement the missing uncertainty in DNN.

In recent years, the emerging nanotechnology device and crossbar structure based neuromorphic computing chips have developed a lot Fuller et al. (2019); Boybat et al. (2018); Yao et al. (2017). The Ohms law and Kirchhoffs law make the crossbar structure very efficient when doing the vector-matrix multiplication (VMM), and the emerging nanoscale nonvolatile memory (NVM) device at each cross point provides additional storage capability (Figure 1). The crossbar holds the devices conductances as memory in peacetime, and performs the computing function when applied voltages. The so-called computing in memory (CIM) architecture can relieve the memory bottleneck, which is the most serious problem in the von Neumann architecture, and make the neuromorphic computing chips more energy and area efficiency. Therefore, the neuromorphic computing has become a promising approach to realize the AI applications, which is full of VMMs and great memory requirement. Besides the energy and area efficiency, the uncertainty is also an important and intrinsic feature of the neuromorphic computing chips and is not well utilized.

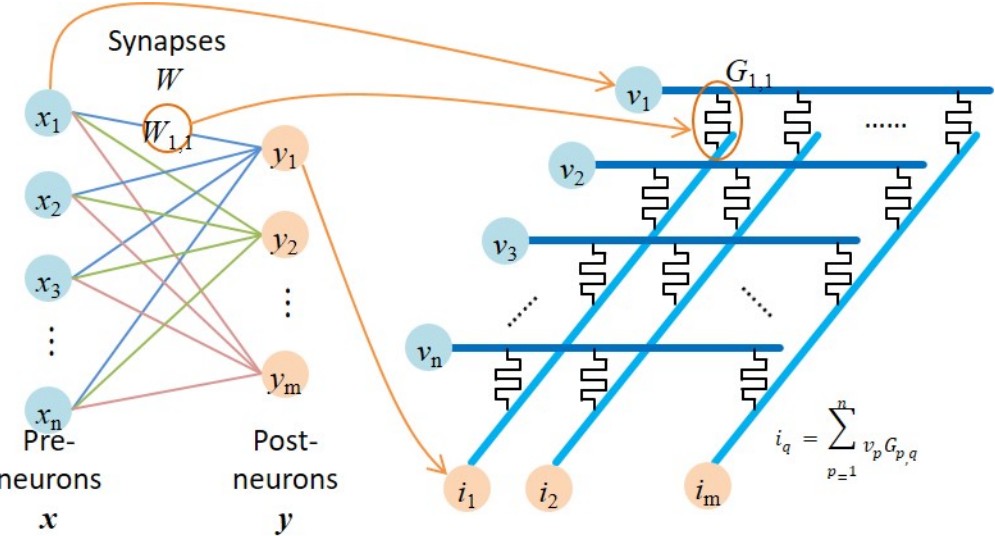

Figure 1: The crossbar structure. V is the applied voltage that correspond to the input x, G is the conductance of devices that correspond to the weight W, I is the output current, which can indicates the output y according to the Ohms law and Kirchhoffs law.

The uncertainty in the neuromorphic computing chips comes from two aspects. The fuzziness is mainly caused by the analog to digital converters (ADCs) and the stochasticity is mainly induced by the NVM devices. According to the Kirchhoffs law, the VMM result is indicated as the summarization of the currents, which is an analog output. It is necessary to use the ADC to convert the analog currents to digital voltages for data transferring. The function of ADC is similar as the activation quantization in the network. The stochasticity of the NVM device is due to the intrinsic physical mechanism Zhao et al. (2017); Lin et al. (2018). The random movement of the particles in the device makes the conductance varied. The output current will be different even applying the same voltage. The stochasticity of the device is usually simulated as a non-ideal factor that makes the network perform worse Prezioso et al. (2015); Ambrogio et al. (2018); Tang et al. (2017). In this work, we proposed a training scheme that utilizes the stochasticity to improve the performance of the neuromorphic computing chips.

## 2 METHODOLOGY

### 2.1 MODELING THE DEVICE STOCHASTICITY

There are several varieties of NVM devices including phase change memories Suri et al. (2012), filamentary migrating oxide devices Wong & Salahuddin (2015), ferroelectric tunnel junction synapses Chanthbouala et al. (2012), and so on. The stochasticity of each types of devices is different due to the different intrinsic physical mechanisms. Without loss of generality, we used the Gaussian distribution to model the device stochasticity. Although the Gaussian distribution may not fit the stochasticity of each types of devices well, it has the basic characteristics, that is, a stable state and the farther away from the stable state, the probability is lower. The mean of the Gaussian distribution is the conductance value of the stable state. The variance of the Gaussian distribution is usually corresponded to the mean according to experimental results Zhao et al. (2017). It is also hard to use a singular model the relations between the mean and the variance of various devices. For simplify and without loss of generality, we assumed that the standard deviation of the distribution is linearly and positively correlated to the mean. Furthermore, the real conductance of device is a positive value. Therefore, the conductances sampled from the distribution which are less than the given value $G_{min}$ will be cut off to $G_{min}$, where $G_{min} > 0$ denotes the minimum conductance that all devices could reach. The model of the devices stochasticity is as follow:

$$G_s = max\left(G_{min}, G'\right), G' \sim N\left(G_0, \alpha G_0^2\right), \tag{1}$$

where $G_s$ denotes the stochastic conductance, $G'$ denotes the value sampled from the Gaussian distribution $N(\mu, \sigma^2)$ with the a mean of $\mu$ and a variance of $\sigma^2$, $G_0$ denotes the conductance of the device when it is in the stable state, and $\alpha > 0$ denotes the level of the devices stochasticity.

## 2.2 Modeling the device fuzziness

The device fuzziness is a by-product of the writing process. Writing the conductance of each device in the crossbar is an essential step when using neuromorphic computing chip to realize an AI application. Before writing, we need to determine the target conductance of each device according to the weights of the neural network. That is what the mapping process do. Besides scaling the weights into the working range of devices conductance $[G_{low}, G_{high}]$, we used the difference of two devices conductances $G_{pos} - G_{neg}$ to express one weight $w$, which can be positive or negative. In order to achieve higher energy efficiency, it is better to use lower conductances to express the same weight. Moreover, in order to make full use of the entire conductance working range, the mapping algorithm is as follow:

$$G_{pos} = \frac{\left(|w| + w\right)\left(G_{high} - G_{low}\right)}{2max_w|w|} + G_{low} \tag{2}$$

$$G_{neg} = \frac{\left(|w| - w\right)\left(G_{high} - G_{low}\right)}{2max_w|w|} + G_{low} \tag{3}$$

where $|w|$ denotes the absolute value of a singular $w$, $max_w|w|$ denotes the maximum of all the $|w|$, $G_{pos}$ and $G_{neg}$ are the target conductances that we want to write. However, the conductance cannot be written accurately due to the stochasticity of the device and the fuzziness of the circuit. There is a variation when manipulate the conductance of the device and the measurement of the conductance is not accurate. The conductance value is obtained by dividing the read current by the applied voltage, which is obviously affected by the stochasticity of the device and the precision of the ADC. Therefore, we need a model to describe the fuzziness. Without loss of the generality, we also used the Gaussian distribution, which is as follow:

$$G_f = max\left(G_{min}, G''\right), G'' \sim N\left(G_{target}, \beta\right), \tag{4}$$

where $G_f$ denotes fuzzy target conductance, $G''$ denotes the value sampled from the Gaussian distribution $N\left(\mu, \sigma^2\right)$ with the a mean of $\mu$ and a variance of $\sigma^2$, $G_{target}$ denotes the target conductance determined in the mapping process, and $\beta$ denotes the level of devices fuzziness, which is correspond to the level of the devices stochasticity, the precision of the ADC, and the writing strategy. For the sake of simplicity, we assumed $\beta$ a constant in this work.

## 2.3 The uncertainty adaptation training scheme

If we program a well-trained DNN into a neuromorphic computing chip directly, the uncertainty will make the performance of the DNN worse, such as decrease the classification accuracy (Tang et al. (2017), Fig. 2). However, the decrease of accuracy can be alleviated by using the proposed uncertainty adaptation training scheme (UATS), and in some cases the accuracy will improved.

The core idea of UATS is to tell the uncertainty to the neural networks during training process and guide them to learn how to deal with this situation. The stochasticity model is introduced in every feed forward (FF) process. When a weight $w$ participates in the calculation of the FF process, we use a sample of random variable $w_s$ to do the calculation instead of $w$. $w_s$ is defined as:

$$w_s = \frac{max_w|w|}{G_{high} - G_{low}}\left(G_{ps} - G_{ns}\right) \tag{5}$$

Both $G_{ps}$ and $G_{ns}$ are obtained by the stochasticity model (1), and the conductances of the stable states $G_{p0}$ and $G_{n0}$ are calculated by (2) and (3) according to $w$, respectively. Actually, the $w_s$ can be approximated as (6) if $G'$ is significantly larger than $G_{min}$.

$$w_s \sim N\left(w, \alpha w^2 + c\right) \tag{6}$$

where $c = 2\alpha \left(\frac{G_{low} \cdot max_w |w|}{G_{high} - G_{low}}\right)^2$.

The fuzziness model is introduced during the training process. After every $k$ epochs of training, every weight $w$ are replaced by a sample of random variable $w_f$, which is defined as:

$$w_f = \frac{max_w |w|}{G_{high} - G_{low}} \left(G_p - G_n\right) \tag{7}$$

Both $G_{pf}$ and $G_{nf}$ are obtained by the fuzziness model (4), and the target conductances $G_{ptarget}$ and $G_{ntarget}$ are calculated by (2) and (3) according to $w$, respectively. Actually, the $w_f$ can be approximated as (8) if $G''$ is significantly larger than $G_{min}$.

$$w_f \sim N\left(w, \beta \frac{max_w |w|}{G_{high} - G_{low}}\right), \tag{8}$$

Besides making the network train in an uncertain way, we also tried to teach it a better way to evaluate the networks performance when there is uncertainty, that is, how to calculate the loss function. The loss function is not calculated by the output of one FF process, but the average output of $n$ FF processes with the same input batch.

## 3 EXPERIMENTS

We evaluated the ideas of UATS on multiple models and datasets.

### 3.1 EXPERIMENT RESULTS ON MNIST

We first investigated the effect of the uncertainty without UATS on MNIST dataset LeCun et al. (1998). Two multilayer perceptron (MLP) models ($28 \times 28$–100–10 and $28 \times 28$–300–10) and a convolutional neural network (CNN) models (LeNet–5) LeCun et al. (1998) are used. There are $60,000$ images in the training set of the MNIST dataset. We randomly selected $50,000$ images for training, and the other $10,000$ images for validation. The $10,000$ images in the test set were used to calculate the test error. $G_{min} = 1\mu S, G_{low} = 5\mu S, G_{high} = 50\mu S$ were used in the experiments. The models were trained in a normal way and then tested with different level of the uncertainty. The fuzziness model (8) was first used to generate the weights that is actually written in the chip, and then the stochasticity model (6) was used to simulate the read variation. We ran 20 trials for every model with every uncertainty level. The average test error and the standard deviation were reported.

As shown in Figure 2, without using the UATS, the uncertainty makes the test errors of both MLP and CNN models higher. The higher level of the uncertainty, the higher the test error. The CNN model (LeNet–5) has the best performance without the uncertainty, but it is also the most affected by the uncertainty (t–test, $p < 0.01$). It is because the average width of LeNet–5 is smaller than the two MLP models, as the 'mlp2' model is more robust to the uncertainty than the 'mlp1'.

Then we validated the power of UATS. We used UATS to tune the weight of the pre-trained model and also retrained the models from the initial. k=5,n=5 were used in the fine-tuning experiment. The number of epochs is 25. $k = 10$, $n = 5$ were used in the retraining experiment and the number of epochs is 100.

As Figure 3 shown, UATS can significantly improve the accuracies with the same level of uncertainty in both retraining and fine-tuning experiments. Most of the retraining results is better than the fine-tuning us UATS. When the uncertainty level is small ($\alpha = 0.1$, $\beta = 0.1$), the UATS achieved a comparable results to the ideal case.

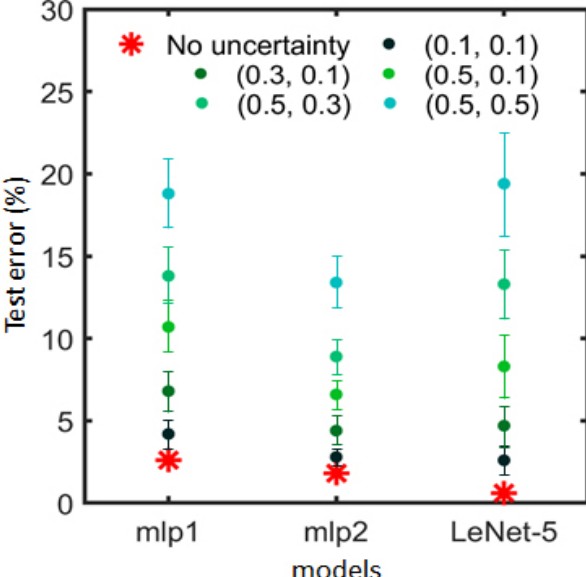

Figure 2: The test errors of different models on MNIST with varies level of uncertainty. The 'mlp1' indicates the smaller MLP model, which has 100 hidden units. The 'mlp2' model indicates the bigger MLP model, which has 300 hidden units. The $(\alpha, \beta)$ indicates the level of uncertainty in the equation (1) and (4), respectively

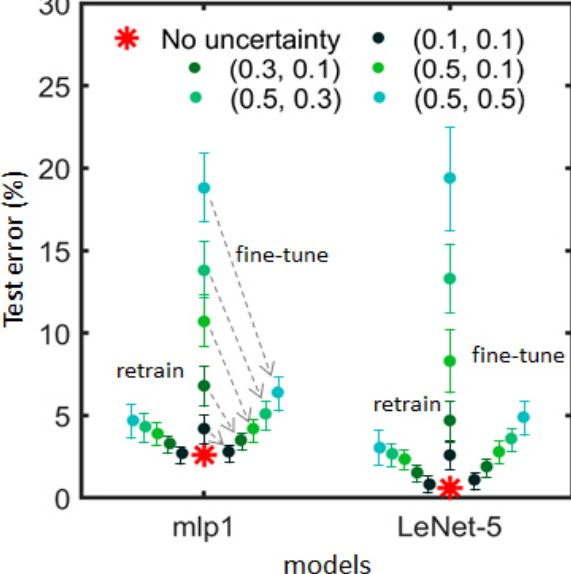

Figure 3: The test errors of different models on MNIST with varies level of uncertainty using UATS. The 'mlp1' indicates the MLP model has 100 hidden units. The $(\alpha, \beta)$ indicates the level of uncertainty in the equation (1) and (4), respectively. The results located to the left of the model results without UATS were obtained by retraining and the right were obtained by fine-tuning.

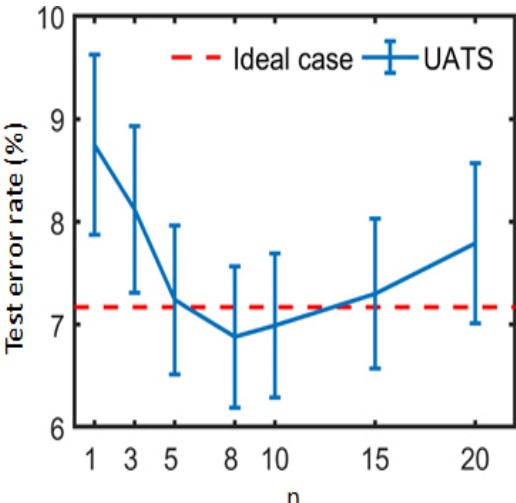

Figure 4: The test error rates of ResNet–44 on CIFAR–10 using UATS with the uncertainty level of $\alpha = 0.1$ and $\beta = 0$. The ideal case indicates the error rate of ResNet-44 model without uncertainty. The n indicates how many FF processes are summarized to calculate the loss function.

## 3.2 EXPERIMENT RESULTS ON CIFAR–10

We also validated the power of UATS on CIFAR–10 dataset Krizhevsky & Hinton (2009) with a more complicated DNN model ResNet–44 He et al. (2016). All the training setting was same as the previous work He et al. (2016) except we used the UATS from the beginning. $\alpha = 0.1$, $\beta = 0.1$, $k = 10$ were used in these experiments. The results shows that UATS can even achieve a lower error rate than the ideal cases with proper hyper-parameters (Figure 4). The UATS performs better when the neural network has more layers. It can be seen as a regularization methods that make the training of DNN easier.

## 4 DISCUSSION

The uncertainty is very important in the intelligent system. The Bayesian network is a very useful method to build an uncertain neural network. However, it usually requires that the distribution of each weight is controllable. This is hard to be realized by the neuromorphic computing chip due to the distribution is determined by the devices. Although there may be some methods to manipulate the conductance distribution of the device, it is not as convenient as UATS, which has no additional circuit required.

We have tried a series of distributions to model the device stochasticity besides the Gaussian distribution, such as the Laplacian distribution, the uniform distribution, and the asymmetrical distributions, such as the lognormal distribution, the asymmetric Laplacian distribution, and the Bernoulli distribution for devices that have two stable states or the random telegraph noise (RTN). Although the modeled behavior of the device from different distributions is significantly different, the performance of network using each type of distribution with the same mean and variance is similar. It is because the VMM transform the individual distribution of each device to a summarization of a large number of random parameters.p

The computation intension of UATS may be a little strong due to the requirement of a large number of random numbers. There are some methods to reduce the requirement of random numbers. Such as samples the weight for every input or every batch instead of the every VMM and using the uncertainty model of VMM results instead of the weights. The simulation speed can be accelerated and achieve similar results.

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
