# OpenReview forum: "A Training Scheme for the Uncertain Neuromorphic Computing Chips"
_ICLR.cc/2020/Conference — Reject_

### Official Review · AnonReviewer2 · 2019-10-24
**Official Blind Review #2**

**Rating:** 1

**Review:**

The authors propose an "uncertainty adaptation training scheme" (UATS) that describes the uncertainty of the neural network in the training process. The authors present experimental results on MNIST and CIFAR-10 demonstrating the utility of their approach.

Overall the quality of the presentation and the exposition in the paper is poor. I am also not convinced about the novelty and importance of this work.  Calibrating neural network uncertainty has been explored quite thoroughly in the bayesian neural nets community - I do not see comparisons with existing work on this subject or justification/explanation of why this work is better than other prior work on this topic.

**Experience Assessment:**

I do not know much about this area.

**Review Assessment: Checking Correctness Of Derivations And Theory:**

I assessed the sensibility of the derivations and theory.

**Review Assessment: Checking Correctness Of Experiments:**

I did not assess the experiments.

**Review Assessment: Thoroughness In Paper Reading:**

I made a quick assessment of this paper.

---

> ### Author Response · Authors · 2019-11-14
> **Response to Reviewer 2**
>
> In Bayesian neural network, the uncertainty of the weight, such as the standard deviation, is usually a trainable parameter. However, in neuromorphic computing chip, this is a fixed restriction, which is determined by the device or the circuit. Our method is to deal with this underlying uncertainties.

---

### Official Review · AnonReviewer3 · 2019-10-24
**Official Blind Review #3**

**Rating:** 6

**Review:**

This paper proposes a way of training neural nets on analog-circuit based chips, which are cursed with uncertainties. Such uncertainties are deeply rooted in the way neural nets are implemented on such chips. Take [c = a x b] as an example. In order to perform this operation, one can set the electric potential to a and the conductance to b and c will be the output current. The problem here is that we cannot set the conductance precisely, which often encodes the weights of a neural net. This implies we cannot precisely program a neural net into these chips. This paper proposes to train a neural net with the presence of such noise, by treating weight as a random variable during training. The experimental results based on simulation suggest this is a much better strategy than programming a neural net into chips imprecisely.

Overall, this paper touches upon an important research problem towards running neural nets on neuromorphic computing chips, which is how to deal with the underlying uncertainties. The proposed algorithm is reasonable and the experimental results look encouraging. However, I would like to ask a few clarification questions. Given authors’ response, I will be willing to adjust my score.

(1) For the baseline, have you tried randomly jittering the network weights after every training iteration in a way that is “blind” to the source of the uncertainties (i.e. conductance)? I would like to understand in what degree modeling the noise helps. If this works out, it implies, (1) we do not have to pay much cost in sampling; (2) there is a simpler way to train neural nets that behave robust when deploying onto neuromorphic computing chips despite the uncertainties.

(2) Is there any intuition behind replacing every weight after every k epochs with new samples (Sec. 2.3)?

(3) The paper does not mention the overhead of estimating the loss with n feed-forward passes dramatically slows the training process. I assume it will slow down training by n times?

(4) There is a comparison between retraining and fine-tuning. Despite being less accurate, is fine-tuning faster to train in terms of the actual training time?

(5) Here I quote the paper “The UATS performs better when the neural network has more layers” (Sec. 3.2). I cannot find an empirical comparison that supports this claim.


**Experience Assessment:**

I do not know much about this area.

**Review Assessment: Checking Correctness Of Derivations And Theory:**

I carefully checked the derivations and theory.

**Review Assessment: Checking Correctness Of Experiments:**

I carefully checked the experiments.

**Review Assessment: Thoroughness In Paper Reading:**

I read the paper thoroughly.

---

> ### Author Response · Authors · 2019-11-14
> **Response to Reviewer 3**
>
> Thanks for the acknowledgement of our work. We believe that the neuromorphic computing is a promising way to introduce AI into people’s daily life with its high energy efficiency. We hope that this work can pave a way forward to improve the performance of the neural network on the neuromorphic computing chips. The responses for the questions is as follows.
>
> (1)	We have tried randomly jittering the weights after every iteration. The results on CIFAR-10 are better than the baseline, but worse than the UATS’s result with n=1 shown in Figure 4 even with a carefully tuned jitter amplitude. Furthermore, if we training the neural networks on the neuromorphic chips, there is no additional cost in sampling the weights.
> (2)	The replacement simulates the weight transfer to the neuromorphic computing chips. The larger k, the higher training speed of the algorithm, but the worse the adaptability to the chip.
> (3)	It will slow down the training. However, this is a onetime cost. And we plan to improve the training speed in the future work.
> (4)	The fine-tuning takes much less epochs to achieve similar accuracy than retraining. Thus, it’s faster.
> (5)	We have compared the accuracies on CIFAR-10 with ResNets that have different layers (20, 44, and 110 layers). The UATS performs better when comparing the 20-layer network and the 44-layer network. However, we didn’t obtain the result of the 110-layer network in time. Thus, we omitted that part of results but forgot to omit this sentence.

---

### Official Review · AnonReviewer1 · 2019-11-04
**Official Blind Review #1**

**Rating:** 1

**Review:**

The paper is hard to read and there are syntactic errors as well as  issues with the grammar. The paper is not at all well written and the contributions very questionable. The paper lacks a conclusion where the main contribution are mentioned and backup. Figures and illustrations are not the best.  From my point of view, this paper is a clear rejection.

I would encourage the authors to be explicit about their contribution and the intellectual products of this work. In addition, I would encourage them the identify comparable methods if any and explicitly enumerate the advantages of their approach against prior work/methods.

**Experience Assessment:**

I have read many papers in this area.

**Review Assessment: Checking Correctness Of Derivations And Theory:**

I assessed the sensibility of the derivations and theory.

**Review Assessment: Checking Correctness Of Experiments:**

I assessed the sensibility of the experiments.

**Review Assessment: Thoroughness In Paper Reading:**

I read the paper at least twice and used my best judgement in assessing the paper.

---

### Decision · Program_Chairs · 2019-12-19

**Decision:**

Reject

**Comment:**

The paper is proposing uncertainty of the NN’s in the training process on analog-circuits based chips. As one reviewer emphasized, the paper addresses important and unique research problem to run NN on chips. Unfortunately, a few issues are raised by reviewers including presentation, novelly and experiments. This might be partially be mitigated by 1) writing motivation/intro in most lay person possible way 2) give easy contrast to normal NN (on computers) to emphasize the unique and interesting challenges in this setting. We encourage authors to take a few cycles of edition, and hope this paper to see the light soon.